# Influence of Landscape Structures on Water Quality at Multiple Temporal and Spatial Scales: A Case Study of Wujiang River Watershed in Guizhou

**Guoyu Xu [1,2], Xiaodong Ren [1,2,*], Zhenhua Yang [3], Haifei Long [4] and Jie Xiao [1,2]**

1    Institute of Karst Guizhou Normal University, Guiyang 550001, China; xgysos@163.com (G.X.); mhxj47@163.com (J.X.)
2    National Engineering and Technological Research Center of Karst Rocky Desertification Prevention and Control, Guiyang 550001, China
3    Department of Water Resources and Environment, Sun Yat-Sen University, Guangzhou 510000, China; yangzh750@sina.com
4    Guizhou Hydrology and Water Resources Bureau, Guizhou 550001, China; longhaifei110@163.com
*    Correspondence: renxiaodong@hotmail.com; Tel.: +86-851-86754106

**Abstract:** Water quality is highly influenced by the composition and configuration of landscape structure, and regulated by various spatiotemporal factors. Using the Wujiang river watershed as a case study, this research assesses the influence of landscape metrics—including composition and spatial configuration—on river water quality. An understanding of the relationship between landscape metrics and water quality can be used to improve water contamination predictability and provide restoration and management strategies. For this study, eight water quality variables were collected from 32 sampling sites from 2014 through 2017. Water quality variables included nutrient pollutant indicators ammoniacal nitrogen ($NH_3$-N), nitrogen ($NO_3^-$), and total phosphate (TP), as well as oxygen-consuming organic matter indicators COD (chemical oxygen demand), biochemical oxygen demand ($BOD_5$), dissolved oxygen (DO), and potassium permanganate index ($COD_{Mn}$). Partial least squares (PLS) regression was used to quantitatively analyze the influence of landscape metrics on water quality at five buffer zone scales (extending 3, 6, 9, 12, and 15 km from the sample site) in the Wujiang river watershed. Results revealed that water quality is affected by landscape composition, landscape configuration, and precipitation. During the dry season, landscape metrics at both landscape and class levels predicted organic matter at the five buffer zone scales. During the wet season, only class-level landscape metrics predicted water contaminants, including organic matter and nutrients, at the middle three of five buffer scales. We identified the following important indicators of water quality degradation: percent of landscape, edge density, and aggregation index for built-up land; aggregation index for water; CONTAGION; COHESION; and landscape shape index. These results suggest that pollution can be mitigated by reducing natural landscape composition fragmentation, increasing the connectedness of region rivers, and minimizing human disturbance of landscape structures in the watershed area.

**Keywords:** landscape metric; water quality; buffer zone; PLSR; Wujiang river watershed

---

## 1. Introduction

Water quality plays a pivotal role in ecosystems, public health, and socio-economic sustainability [1–3]. Urbanization and economic development have put significant stress on water systems across the world, making water quality an issue of global importance [4,5]. Land use/cover change (LUCC) is a complex process subjected to interactions between social systems and the natural

environment at different temporal and spatial scales [6,7]. As IGBP (International Geosphere-Biosphere Program) and LUCC projects are implemented worldwide, research on the effects of changes in landscape patterns on water environments has become a trending topic in resource management and environmental science fields [8–10]. Water quality of watersheds is not only regulated by natural processes such as rainfall, soil types, and landform, but also by anthropogenic activities. In addition, land use and landscape structures also affect hydrological processes and water quality [11–13]. The relationship between water quality of rivers and riparian land use are subject to spatiotemporal variation [14–16]. The development of GIS has helped advanced both quantitative and qualitative analysis tools for understanding the relationship between land structure and water quality, and have contributed greatly to watershed planning and management fields [17–23].

Landscape structures are based on land use composition and landscape configuration. The latter refers to the physical distribution or spatial characteristics of patches at landscape and class levels. Class-level metrics describe the amount and spatial distribution of patches belonging to a particular land use class; landscape-level metrics describe the spatial pattern of the entire landscape mosaic [24,25]. Landscape configuration is an important factor for determining the relationships between land use and water quality in adjacent aquatic systems [9,21,26–28]. However, there exists discrepancy among researchers regarding which aspects of landscape pattern should be analyzed. A number of studies have been conducted on landscape structures at a landscape level, while very few studies have attempted to explore spatial characteristics for each patch at the class level [15,18,19,21,26]. To better evaluate the complexity of landscape metrics on water quality, more studies at both class and landscape levels are needed. This study attempts to help fill this gap in the literature.

Buffer zones can greatly mitigate denitrification and the deposition and absorption of contaminants from non-point pollution into adjacent water [14,29–37]. The mitigation effects of the buffer zone vary based on its proximity to the water source. As such, the relationship between water quality indicators and landscape metrics are scale-sensitive [30–33]. Studies have highlighted land use/landscape within different buffer zones scales. These studies conclude that changes in the spatial characteristics of landscape patterns are a leading factor impacting water quality deterioration [32,34,38,39], the extent and scale of which are case-specific. Research also indicates there is a substantial relationship between precipitation and water quality, as precipitation and surface runoff exert significant impact on flow rates and concentration of contaminants in stream water. Such variations are seasonal [1,33,34,40]. Li et al. found that riparian land use exerted the largest influence on the upper Han River basin of China during the dry season [39]. However, Shen et al. found that landscape characteristics had a strong correlation with water quality during the rainy season [26,31]. Exploring the relationships between watershed landscape characteristics and seasonal variability of stream water contaminants bears considerable importance for the optimization of buffer zones and improvement of watershed management [5,31].

Landscape metrics are neither independent variables nor do they exist collinearly. Some indicators can skew statistical analysis and yield potentially misleading results because different landscape indicators may be interrelated [40]. High multicollinearity will cause a poor performance of multivariate regression even if a subset of expression levels is selected [40,41]. Hence, the compounding relationship between water contaminants and landscape metrics remains difficult to examine. Partial least squares regression (PLSR) helps overcome this issue by combining the features of principal component analysis and multiple linear regression. It is also one of the most useful algorithms for prediction when multicollinearity exists and when the number of predictor variables is more than that of observations [42–44]. PLSR is employed in conjunction with variable influence on projection (VIP) to define which dependent variables can effectively explain independent variables [45].

The relationship between the landscape characteristics of a watershed—including land cover, topography, climate, soil, and geology—and water quality is complicated and place-specific [46–48]. While many studies on the influence of landscape structure on regional hydrological processes and water quality have been conducted, there has only been one published study on large watersheds

in karst areas (published in Chinese) [48]. Karst areas, due to their steep slopes and shallow, highly erodible soil, create particularly vulnerable conditions for watersheds. Watersheds in karst areas are especially prone to soil and surface-level pollution runoff as a result of precipitation and human activities. Furthermore, the high population density and proportion of people involved in agricultural production in the Wujiang river watershed has resulted in significant landscape structure changes, as well as changes in water quality.

This study aims to (1) analyze the quantitative relationship between landscape metrics and water quality across scale (five buffer zones at different extents from the sample site) and time (dry and wet seasons) in the Wujiang river watershed, and (2) identify which landscape metrics can effectively predict water quality when applying PLS regression.

## 2. Materials and Methods

### 2.1. Study Area

The Wujiang river watershed (22°07′~30°22′ N, 104°18′~109°22′ E) is the largest watershed in Guizhou province, as shown in Figure 1. The watershed covers a total area of 32,300 km² and runs 889 km in length from west to east. Karst topography and landform are widely developed in this area. The Wujiang river watershed lies in a subtropical monsoon climate. The average annual temperature is approximately 13–18 °C, and average annual rainfall 900–1400 mm. The population of the Wujiang river basin in 2015 was over 21 million, with 61.3% of the population involved in agricultural production. Additionally, 37.47% of the population in the Wujiang river watershed lives in urban areas and are involved in non-agricultural livelihoods. Some districts have a high urbanization rate; for example, Yunyuan district has an urbanization rate of 97.31%. Mining and manufacturing industries are dominant in this district. Watershed pollution sources include industry, urban waste water, agriculture, and large-scale livestock farming.

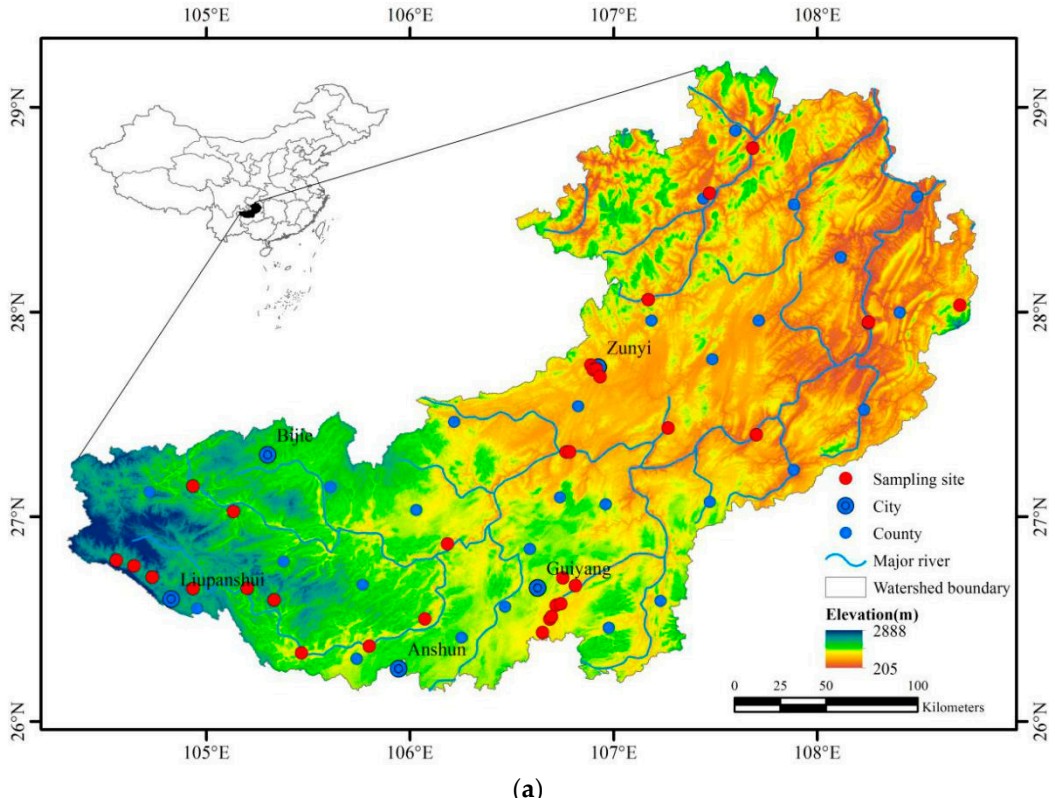

**(a)**

**Figure 1.** *Cont.*

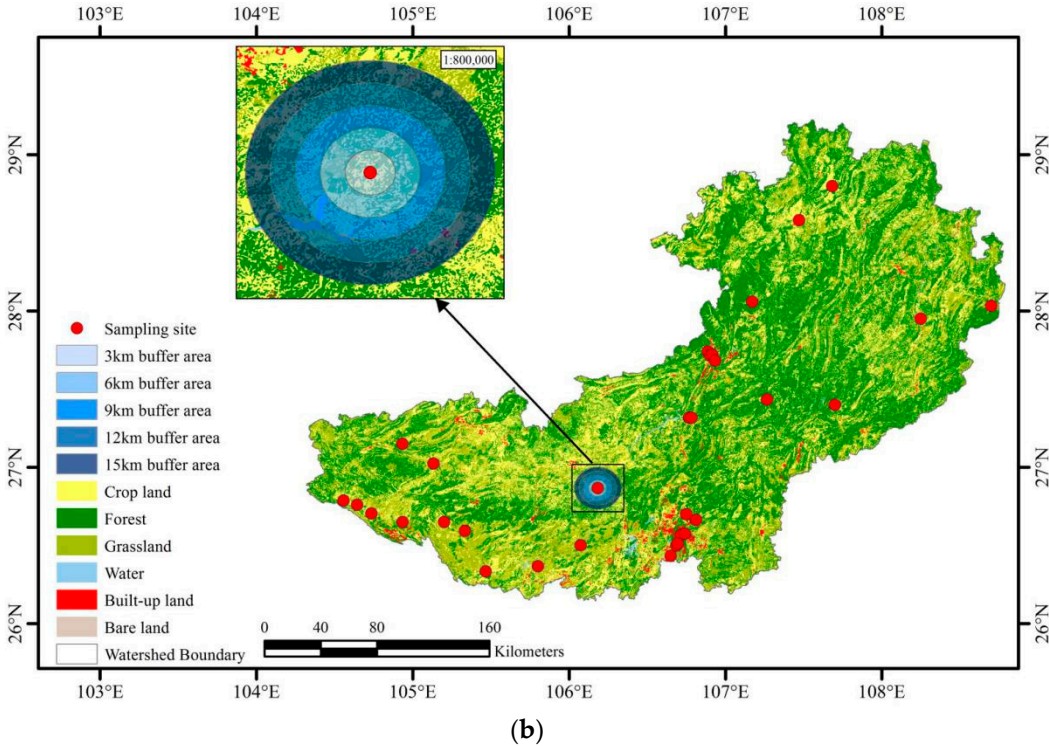

(**b**)

**Figure 1.** Wujiang river watershed displaying (**a**) a digital elevation model and (**b**) land use pattern.

*2.2. Water Sampling and Analysis*

Thirty-two sample sites along the main stream and its tributaries were established by the Guizhou Environmental Monitoring Centre to monitor water contaminants in the watershed. The water quality detection equipment was set up in the outlets of upstream tributaries, as shown in Figure 1. In the field survey, the coordinates of sampling site points were recorded using a portable GPS (Garmin eTrex201x). River water samples were taken at monthly intervals from January to December throughout 2014 to 2017. Eight water quality parameters were chosen to assess chemical characteristics in their corresponding sub-watersheds in accordance with the environmental quality standards for surface water (Chinese State Environmental Protection Bureau, 2002). Ammoniacal nitrogen ($NH_3^+$-N), nitrate ($NO_3^-$), and total phosphate (TP) were selected as indicators of nutrient pollutants; chemical oxygen demand (COD), biochemical oxygen demand ($BOD_5$), dissolved oxygen (DO), and potassium permanganate index ($COD_{Mn}$) were selected as indicators of oxygen-consuming organic matter. These indicators were reported in units of mg/L. Water acidity (pH) was also analyzed. The experiment methods are shown in Table 1. Rainfall data were collected from 20 meteorological stations located within or close to the watershed. The data were obtained from China Meteorological Data Network (http://data.cma.cn/data/cdcindex/cid/8e65c709b3220e70.html). To compare seasonal variation, water quality analysis was split into two seasons periods: the wet season (April through September); and the dry season (November through March).

**Table 1.** Water quality experiment methods.

| | |
|---|---|
| $NH_3^+$-N | spectrophotometric method with salicylic acid; |
| DO | iodine quantity method; |
| $COD_{Mn}$ | acidic (alkaline) potassium permanganate method; |
| COD | dichromate method; |
| $BOD_5$ | dilution and inoculation method |
| $NO_3^-$ | phenol disulfonic acid spectrophotometry |
| TP | ammonium molybdate spectrophotometric method |
| pH | PH-40A portable PH acidity meter |

### 2.3. Spatial Data

The land use pattern data set was provided by Resources and Environmental Sciences Data Center, Chinese Academy of Sciences (RESDC) (http://www.resdc.cn). Data were provided at 30 m × 30 m resolution and included six broad categories of land use type: (1) forest, including woodland and sparsely forested woodland; (2) crop land, including paddy fields and dry lands; (3) grass land, such as thickets and herbs; (4) water areas, such as reservoirs, lakes, and canals; (5) built-up land, such as industrial and residential areas; and (6) bare land, such as gravel, bare ground, and bare rocks. Most sample sites in our study had large catchments, so a maximum of 15 km was used as the buffer zone. Using the Buffer Wizard tool in ArcGIS 10.2, five buffer zones (3, 6, 9, 12, and 15 km) were generated for each of the selected 32 sampling points.

Nine metrics at the landscape level and four metrics at the class level were selected to represent patch size, shape, structure, diversity, and aggregation, as shown in Table 2. These metrics explain the land composition as well as spatial configuration of the landscape. FRAGSTATS 4.2 software was employed to calculate these landscape metrics using four-cell domain (http://www.umass.edu/landeco/research/fragstats/fragstats.html) [25].

### 2.4. Statistical Analysis

A *t*-test was conducted on independent samples to detect seasonal differences in water contamination concentrations. The abnormality of the data was excluded using a Kolmogorov–Smirnov test. Prior to performing PLSR, a preliminary analysis was conducted to test the collinearity of landscape metrics.

PLSR was selected to analyze the impact of landscape metrics on water quality across different buffer zones and seasons. PLS finds a set of orthogonal X-components that maximizes the level of explanation of Y. This method also provides a predictive equation for Y in terms of the Xs. Other information about PLSR is referred to in the literature [49,50].

PLS models for the eight water contaminants (Y) and 33 landscape metrics (X) were constructed and analyzed for correlation. The PLSR was computed for each of the five buffer zones during wet and dry seasons. The appropriate number of significant PLSR components was determined using cross-validation. All the variables were transformed with a natural logarithm to prevent negative values of predictions and to normalize the distribution of each variable. For the models derived using PLS, the variance in the data matrix explained by the first and second principal component were used as a measure of the model's goodness-of-fit. A value above 60% was set as the limit for an acceptable model.

Root mean square error of cross-validation was used to assess the predictive ability of the PLSR model [50]. A coefficient of variable importance for the projection, derived from the sum of the square of the PLSR weights across all components, was used to evaluate the statistical importance of the independent variables for explaining variation in the dependent variable [40,41,43]. Variables with VIP scores greater than 1 are important for explaining the independent variables. Standardized regression coefficients (RC), which indicate the direction and strength of the impact of each variable, were also used in the PLS regression model. All the statistic procedures were implemented in R (R Development Core Team, 2014) (https://www.r-project.org/).

**Table 2.** Landscape metrics at class and landscape levels used in this study.

| Index | Description | Calculation |
|---|---|---|
| Patch density (PD) | Number of patches per unit area (unit: n/100 ha) | $\text{PD} = (n_i/A) \times 10^6$ [a] <br> $\text{PD} = (N/A) \times 10^6$ [b] |
| Edge density (ED) | Total length of all edge segments per hectare for the considered landscape/class (unit: m/ha) | $\text{ED} = (E/A) \times 10^6$ [b] <br> $\text{ED} = \frac{\sum_{k=1}^{m} e_{ik}}{A}$ [a] [b] |
| Aggregation index (AI) | Degree of dispersion of the patches equals the number of like adjacencies involving the corresponding class (unit: %) | $\text{AI} = \left[ \sum_{i=1}^{m} \left( \frac{g_{ij}}{\max \to g_{ij}} \right) p_{ij} \right] 100$ [a] |
| Percent of landscape (PLAND) | Event when the entire image is comprised of a single patch. | $\text{PLAND} = \frac{\sum_{j=1}^{n} a_{ij}}{A} \times 100$ |
| Contagion index (CONTAG) | Tendency of land use types to be aggregated (unit: %) | $\text{CONTAG} = \left[ 1 + \frac{\sum_{i=1}^{m} \sum_{k=1}^{m} [(p_i)(g_{ik}) / \sum_{k=1}^{m} g_{ik}][\ln p_i((g_{ik}) / \sum_{k=1}^{m} g_{ik})]}{2 \ln m} \right] \times 100$ |
| Shannon's diversity index (SHDI) | Patch diversity in landscape (no unit) | $\text{SHDI} = \sum_{i=1}^{m} (p_i \times \ln p_i)$ |
| Landscape shape index (LSI) | Sum of the landscape boundary divided by the square root of the total landscape area (no unit) | $\text{LSI} = \frac{0.25 \sum_{k=1}^{m} e_{ik}^{j}}{\sqrt{A}}$ |
| Interspersion and juxtaposition index (IJI) | Measurement of the extent to which patch types are interspersed. | $\text{IJI} = \frac{\sum_{i=1}^{m} \sum_{k=i+1}^{m} [(e_{ik}/E) \times \ln(e_{ik}/E)]}{\ln[(m(m-1))]} \times 100$ |
| Perimeter-area fractal dimension (PAFRAC) | PAFRAC is meaningful if the log-log relationship between perimeter and area is linear over the full range of patch sizes. | $\text{PAFRAC} = \frac{\left[ n_i \sum_{j=1}^{n} \left( \ln p_{ij} \cdot \ln a_{ij} \right) \right] - \left[ \left( \sum_{j=1}^{n} \ln p_{ij} \right) \left( \sum_{i=1}^{n} \ln a_{ij} \right) \right]}{\left( n_i \sum_{j=1}^{n} \ln p_{ij}^{2} \right) - \left( \sum_{j=1}^{n} \ln p_{ij} \right)^{2}}$ |
| Patch cohesion index (COHESION) | Patch cohesion index measures the physical connectedness of the corresponding patch type. | $\text{COHESION} = \left[ 1 - \frac{\sum_{j=1}^{n} p_{ij}^{*}}{\sum_{j=1}^{n} p_{ij}^{*} \sqrt{a_{ij}^{*}}} \right] \cdot \left[ 1 - \left[ \frac{1}{\sqrt{Z}} \right] \right]^{-1} \cdot (100)$ |

[a] calculated at class level. [b] calculated at landscape level.

## 3. Results

### 3.1. Spatial and Seasonal Variations of Water Contaminant

The *t*-test indicated that water quality indicators in most sampling sites exhibited significant seasonal differences ($p < 0.05$). The mean concentration of the eight water contaminant parameters was mapped using ArcGIS 10.2. The natural break method was adopted to group the contaminant concentrations into five categories [12,26,39]. The seasonal variations in water quality in Wujiang river watershed are exhibited in Figures 2 and 3. The water quality in most sampling sites generally met Grade III water quality standards. However, water quality in six of the sampling sites were considerably affected by human activity. They met the Grade V inferior quality standards. DO, COD, $BOD_5$, $NH_3^+$-N, DO, and TP were considered as the main water pollution indicators in accordance with the environmental quality standards for surface water (GB3838-2002). The majority of sites exhibited significant spatial differences in $BOD_5$, COD, and $NH_3^+$-N concentrations. Seasonal variation of pH was insignificant, while other water quality indicators fluctuated seasonally. The mean concentrations of DO (range: 6.2–8.39 mg/L), TP (range: 0.11–0.23 mg/L), COD (range: 10.15–13.59 mg/L), and $NH_3^+$-N (range: 0.46–1.4 mg/L) were higher during the dry season than the wet season. Their fluctuations were closely connected to rainfall. By contrast, $COD_{Mn}$ (range: 0.8–2.1 mg/L), $BOD_5$ (range: 0.566–2.53 mg/L), and $NO_3^-$ (range: 1.76–2.25 mg/L) were higher during the wet season than the dry season.

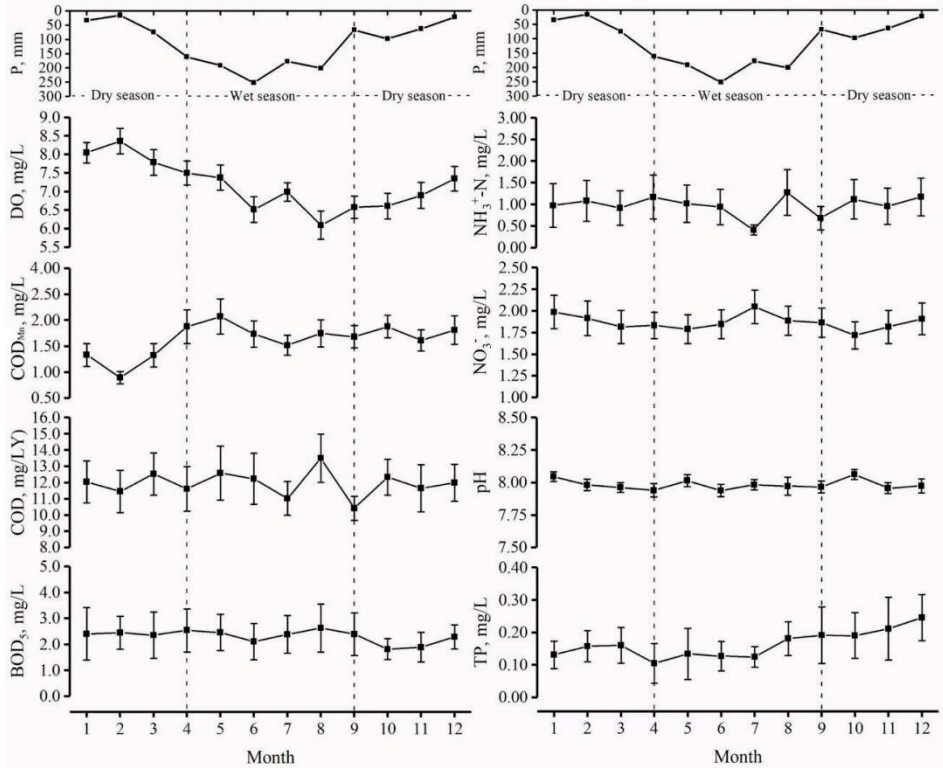

**Figure 2.** Mean values of eight water quality indicators from 2014 to 2017 during wet and dry seasons.

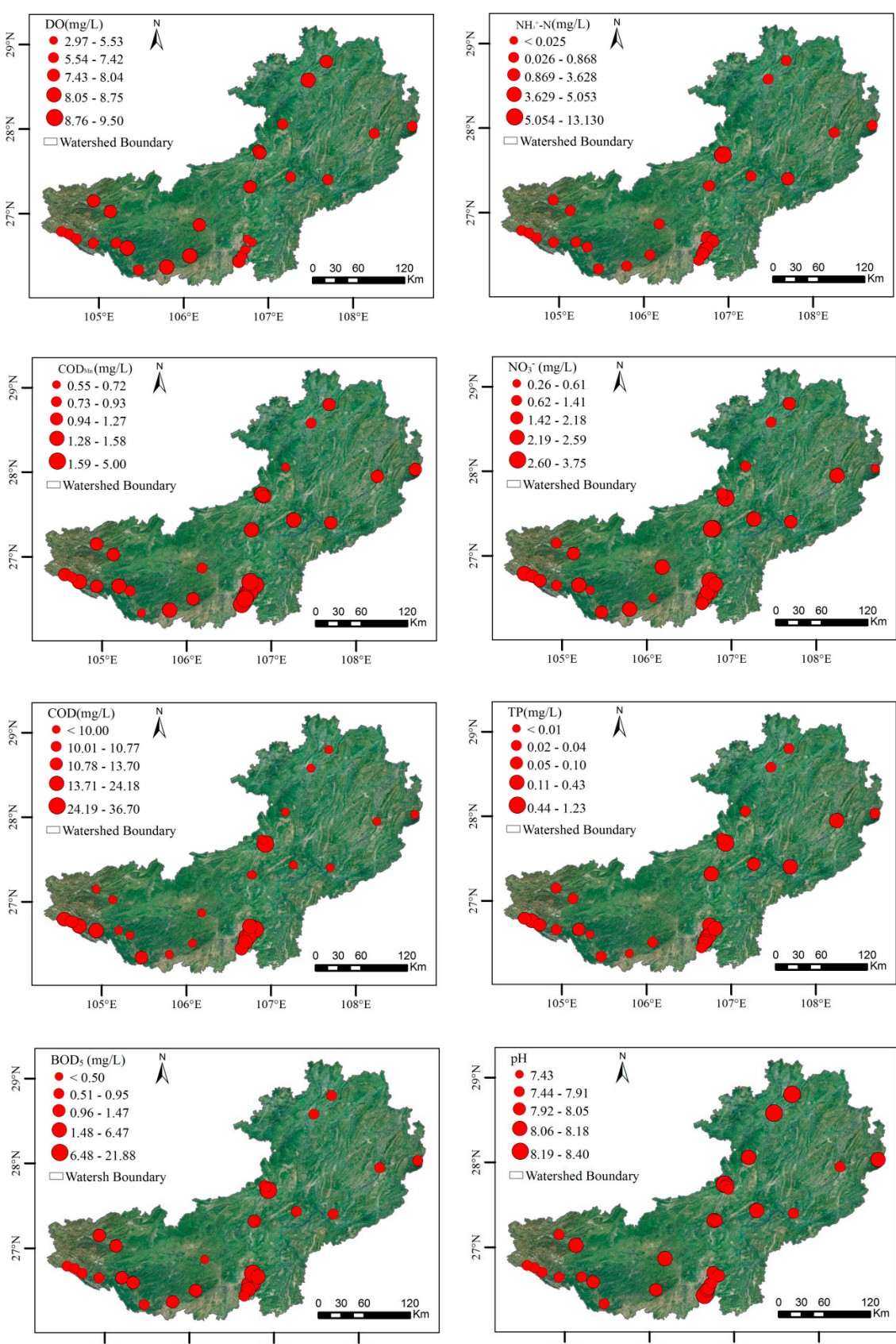

**Figure 3.** Spatial variation of water quality indicators in Wujiang river watershed. Range of values were categorized into five classes using the natural breaks method.

### 3.2. Characteristics of Landscape Composition in Different Buffer Zones

The percentage of land use types at the 32 sampling points varied across the five buffer zones, as shown in Figure 4. Forest, crop land, and grass land were the dominant land cover types. Forest was the largest land use type across all buffer zones, with a mean coverage of 41.93%. As the distance from the sample site increased, the percentage of forest land increased, whereas that of grass and built-up land decreased. The average proportion of crop land was 27.53%, and there was insignificant difference between the mean values across the five buffer zones. Built-up land area within the five buffer zones gradually decreased with distance from the sample site, from a mean of 19.93% in the 3 km buffer zone, to 5.67% in the 15 km buffer zone. The proportion of built-up land in sample sites 17, 18, 29, and 30 exceeded 50% within the 3 km buffer zone. Moreover, the built-up land was unevenly distributed and varied across the five buffer zones. The third main land use type was grass land (22.61% of the total area). The highest proportion of grass land was distributed in the 3 km buffer and decreased with increasing distance from the sample site. Water and bare land occupied a relatively small percentage of the study area and were distributed unevenly across the five buffer zones.

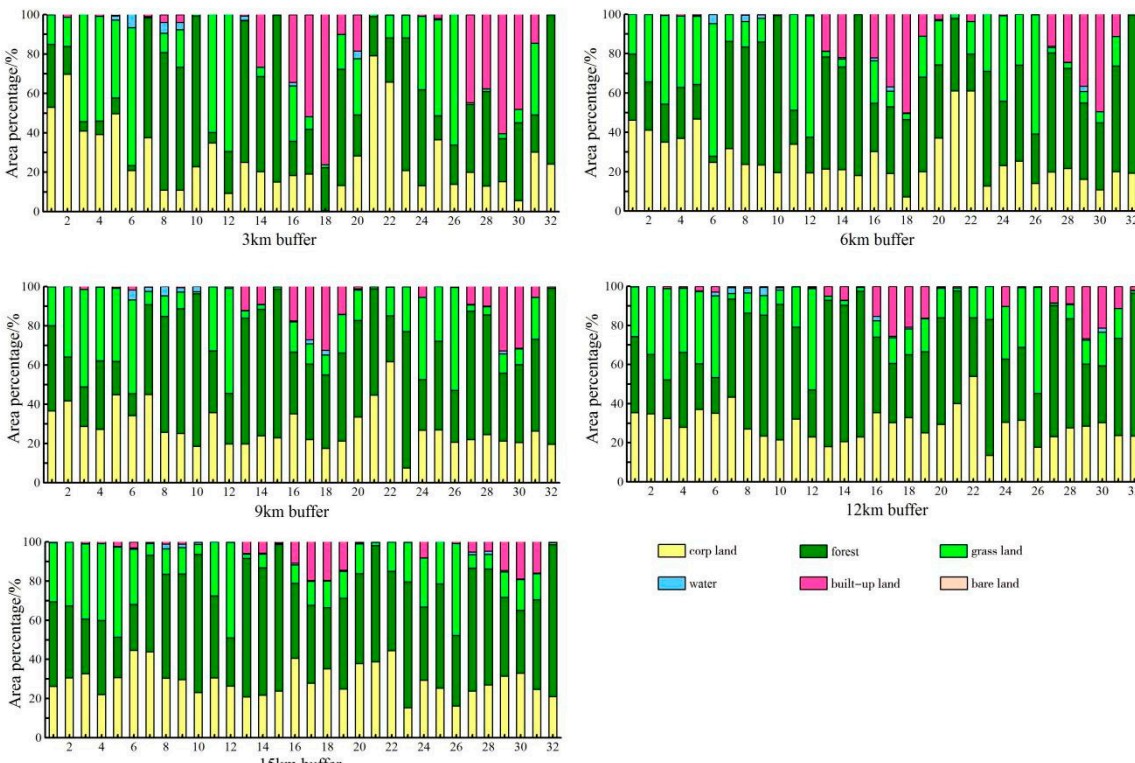

**Figure 4.** Landscape composition characteristics for Wujiang river watershed at five buffer scales.

### 3.3. Characteristics of Landscapes at Class-Level in Different Buffer Zones

The distribution of different landscape metrics at the class level varied across the 32 sampling sites as well as across the five buffer zones, as shown in Figure 5. Across all buffer zones, forest and crop land made up the highest proportion of land use. The value of PLAND (percent of landscape) for forest ranged from 0.418 to 69.73, and PLAND for crop land ranged from 3.101 to 84.67. The value of PLAND for grass land ranged from 0.671 to 77.673, and that for built-up land ranged from 0.024 to 77.202. The value of PLAND for grass land was highest in the 15 km buffer zone. The value of PLAND for built-up land was highest in the 3 km buffer zone. The value of PLAND for water ranged from 0.048 to 48.765 and that of bare land ranged from 0.038 to 0.34.

The PD (patch density) for forest land, built-up land, and grass land increased with increasing distance from the sample sites, while the PD for crop land and bare land decreased with increasing

distance from the sample sites. The PD for water varied across buffer zones. The value of ED (edge density) for crop land ranged from 37.86 to 1.39, forest land from 38.81 to 5.91, and grass land from 48.71 to 9.81. The mean value of ED for built-up land, water, and bare land were 24.19, 0.41, 4.30, respectively.

The AI (aggregation index) for crop land, forest land, and grass land varied across sample sites and buffer zones. The AI for crop land ranged from 85.67 to 99.04, that of forest from 89.77 to 98.23, and that of grass land from 88.51 to 96.79. The value of AI for water, built-up land, and bare land increased with distance from the sample site, as land use types became more diverse.

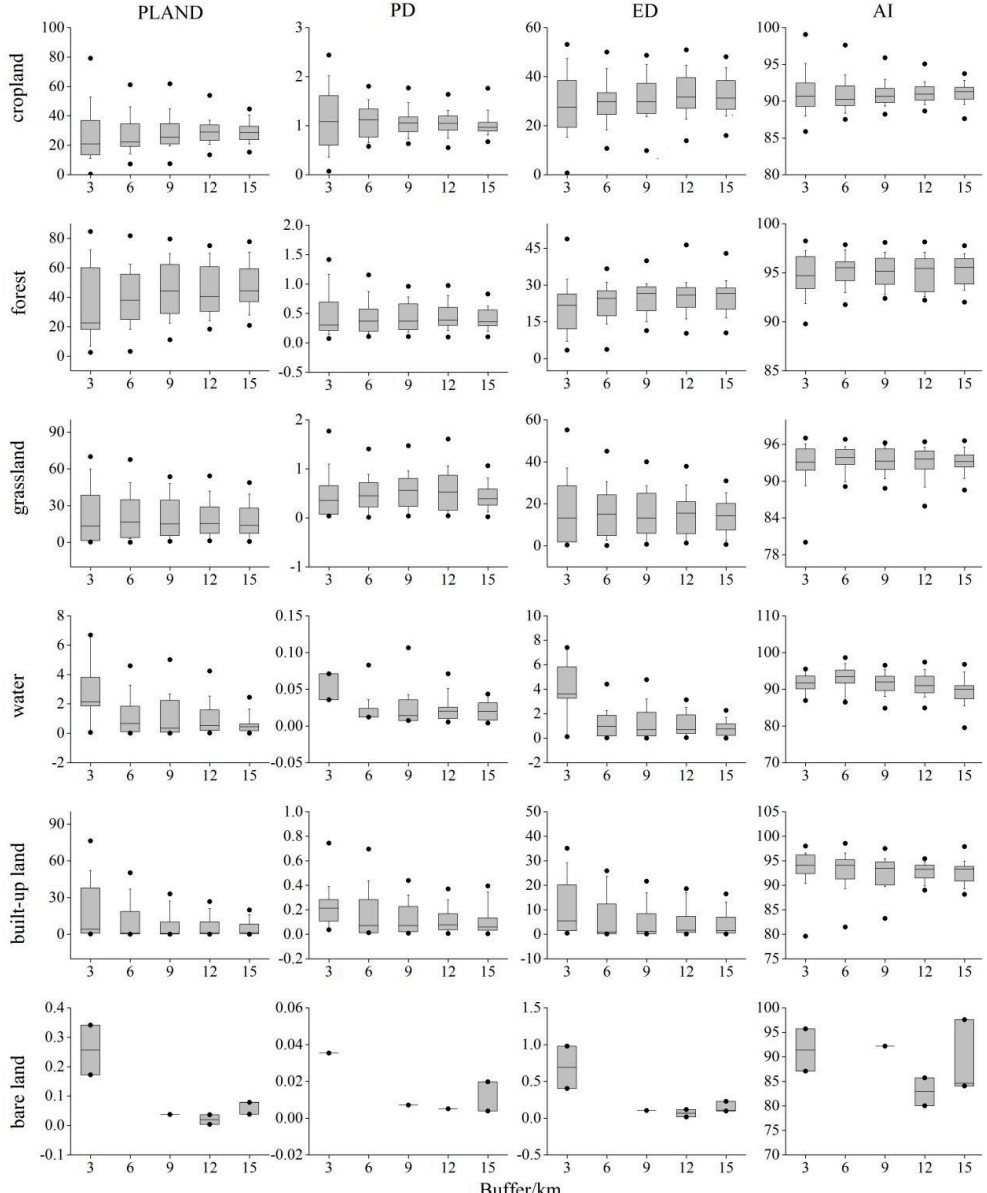

**Figure 5.** Class-level landscape metric indices across buffer zones (displaying value ranges for 32 sampling sites).

*3.4. Characteristics of Landscapes at Landscape Level across Buffer Zones*

Figure 6 displays the value ranges for nine landscape-level metrics at the 32 sample sites across five buffer zones. The value of PD across the five buffer zones ranged from 0.356 to 4.06 n/100 ha. While PD is an indicator of fragmentation, ED and SHDI (Shannon's diversity index) indicate both fragmentation and heterogeneity. The value of ED and SHDI ranged from 18.63 to 64.84, and 0.45 to 1.43 at the 3 km buffer zone, indicating that the highest degree of fragmentation and heterogeneity for

landscape appeared in the 3 km buffer zone. The value of ED and SHDI decreased gradually with increasing distance from the sampling sites. LSI (landscape shape index) and PAFRAC (perimeter-area fractal dimension) reflected the shape of patches. The value of LSI was lowest in the 3 km buffer zone (2.6–9.12), and increased with increasing distance from the sample site, with a value of 8.27–20.22 at the 15 km buffer zone. The value of PAFRAC showed an opposite trend, decreasing as the distance from the sample site increased. CONTAG (contagion index) provides a measure of aggregation, which was greatest at the 6 km buffer zone (42.82–77.62), and lowest in the 12 km buffer zone (38.28–74.25). COHESION (patch cohesion index) presented physical connectedness of patches. The highest value of COHESION (97.062–99.71) appeared at the 9 km buffer zone. LJI (landscape juxtaposition index) represented spatial configuration. For LJI, the value ranged from 84.54 to 97.28 across the five buffer zones. The LPI (largest patch index) values varied widely across the sample sites and buffer zones, from 4.53 to 83.77.

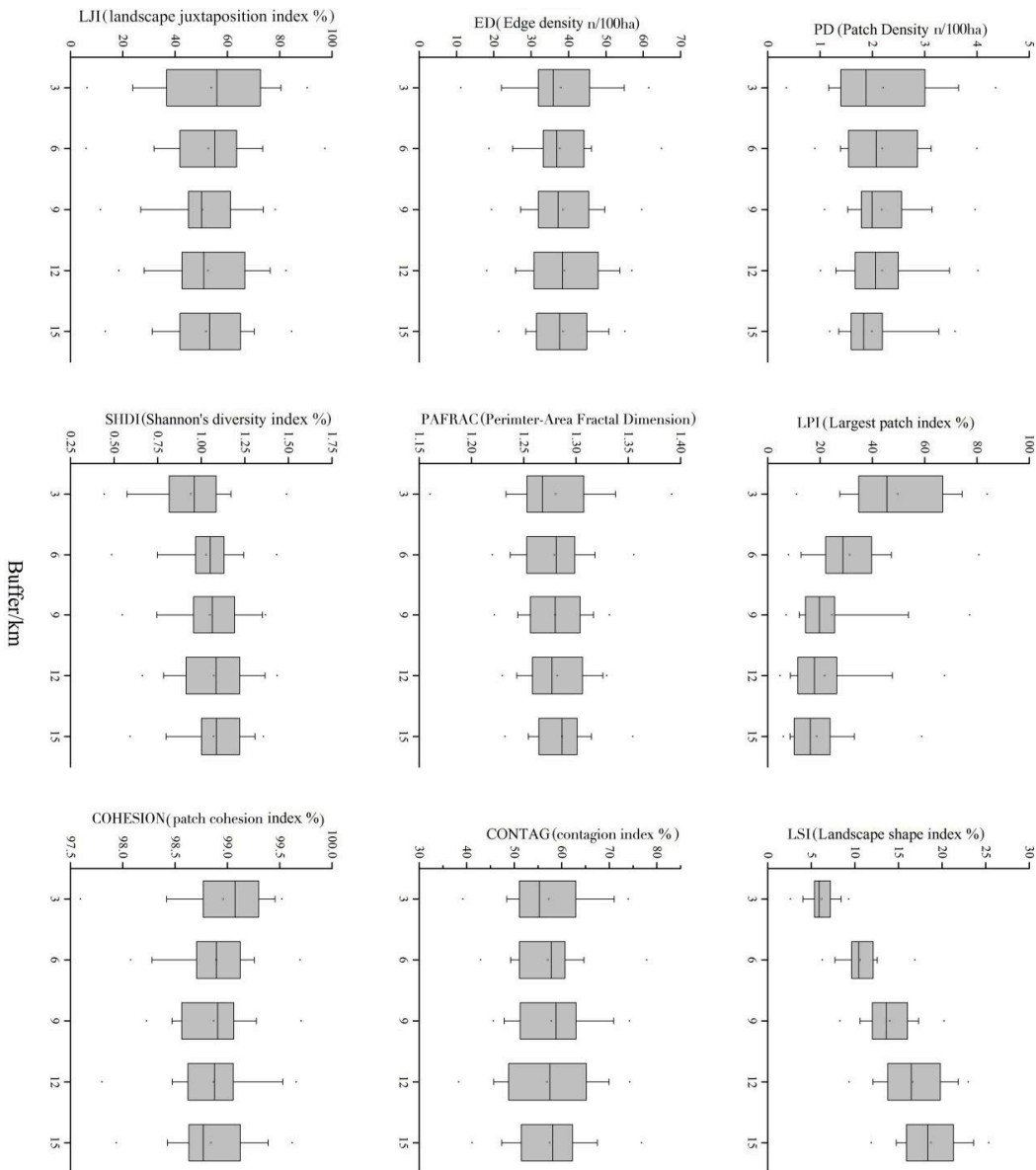

**Figure 6.** Landscape-level metric indices across buffer zones.

### 3.5. Linkages between Water Contaminants and Landscape Metrics across Buffer Zone Scales Based on PLSR

Prior to using PLSR, a Pearson correlation was conducted as a preliminary analysis to show that many of the watershed landscape metrics descriptors are co-linear, as shown in Figure 7. Different principal components were extracted by PLSR in each of the buffer zones. The basis for selecting the number of principal components was that the RMSE (Root Mean Square Error) of each model reached a stable minimum value. The RC and the value for variable importance for the projection (VIP) were a comprehensive expression of the relative importance of the landscape metric variables [38]. Tables 3 and 4 present the coefficients estimated from PLSR models. The PLSR models were used to explore the correlation between water contaminants and landscape metrics.

Water quality degradation could largely be determined by landscape structures, the effects of which varied according to season and buffer zone. During the dry season, as shown in Table 3, landscape metrics at both class and landscape levels correlated with water quality, though to differing degrees. Class-level metrics ED, PLAND, and AI for built-up land were correlated with COD, $NO_3^-$ and TP in the 3, 6, and 12 km buffer zones. ED and AI for forest land were negatively correlated with DO, $COD_{Mn}$, and COD in the 9 and 12 km buffer zone. ED and PD for grass land were negatively correlated with $BOD_5$ and $COD_{Mn}$, respectively, in the 12 and 15 km buffer zones. Landscape-level metrics CONTAG and COHESION were correlated with $BOD_5$, COD, and $COD_{Mn}$ at four of the five buffer zones. ED was negatively correlated with $COD_{Mn}$, COD, and $BOD_5$ in the 3 and 15 km buffer zones. LSI was positively correlated with DO and negatively correlated with COD and $COD_{Mn}$ in the 3 and 15 km buffer zones. LPI was positively correlated with $COD_{Mn}$ in the 9 km buffer zone.

During the wet season, as shown in Table 4, only class-level metrics correlated with water quality indicators. PLAND and ED for crop land were positively correlated with DO and negatively correlated with COD and $BOD_5$ in the 9 km buffer zone. ED and AI for grass land were negatively correlated with $COD_{Mn}$ and COD in the 9 km buffer zone. AI for water was negatively correlated with DO while negatively correlated with $NH_3^+$-N, $COD_{Mn}$, COD, $BOD_5$, and $NO_3^-$ in the 9 km buffer zone. AI for built-up land was positively correlated with DO and negatively correlated with COD and $BOD_5$ in the 6 km buffer zone.

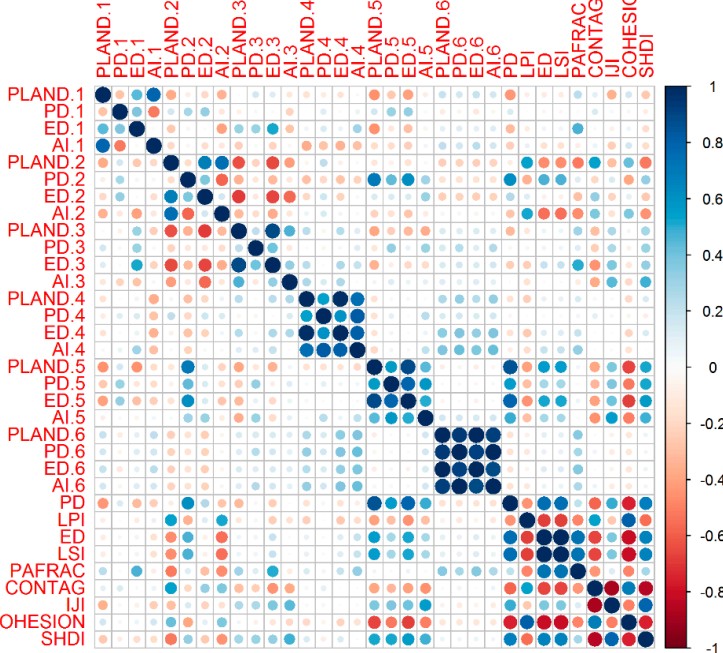

**Figure 7.** Correlation of the landscape metrics used in the Pearson correlation. Note: metric value at class-level category 1 (crop land), 2 (forest), 3 (grass), 4 (water), 5 (built-up land), 6 (bare land).

**Table 3.** Summary of the partial least squares regression (PLSR) models for specific water contaminants during the dry season.

| | | DO | COD$_{Mn}$ | COD | BOD$_5$ | NH$_3^+$-N | NO$_3^-$ | TP |
|---|---|---|---|---|---|---|---|---|
| | ED.5 | | | **−0.098** | | | | |
| | ED [a] | | | **−0.013** | −0.016 | | | |
| 3 km | LSI [a] | | | **−0.050** | | | | |
| | CONTAG [a] | | | | 0.017 | | | |
| | COHESION | | | 0.047 | | | | |
| | PLAND.5 [a] | | | **0.046** | | | **0.031** | |
| 6 km | ED.5 [a] | 0.005 | | −0.031 | | | | |
| | PD | | | −0.003 | | | | |
| | CONTAG [a] | | | 0.027 | | | | |
| | ED.2 [a] | | −0.01 | **−0.064** | | | | |
| | PD | | | −0.004 | | | | |
| 9 km | LPI | | 0.049 | | | | | |
| | CONTAG [a] | | 0.035 | 0.035 | | | | |
| | COHESION [a] | | **0.033** | **0.0326** | | | | |
| | SHDI | | **−0.001** | **−0.0012** | | | | |
| | ED.3 | | | | −0.001 | | | |
| 12 km | AI.5 [a] | | | | | | **0.031** | −0.011 |
| | CONTAG | 0.005 | | | | | | |
| | ED.1 [a] | 0.007 | −0.05 | | | | | |
| | AI.2 [a] | −0.001 | 0.01 | | | | | |
| | PD.3 [a] | | **−0.002** | | | | | |
| | ED.3 | | | | −0.001 | | | |
| 15 km | PD | | −0.003 | | | | | |
| | ED [a] | **0.02** | **−0.061** | | | | −0.006 | |
| | LSI [a] | **0.004** | **−0.024** | | | | −0.002 | |
| | CONTAG [a] | **−0.056** | **0.037** | | | | | |
| | COHESION | **−0.036** | **0.0228** | | | | 0.022 | |

Note: [a] is used to identify the significant independent variables on the basis of VIP (variable influence on projection) scores >1. [b] Bold font indicates the independent variables that passed 95% confidence test ($p < 0.05$), non-bold font represents independent variables that passed 90% significant test ($p < 0.1$).

**Table 4.** Summary of the PLSR models for specific water contaminants during the wet season.

| | | DO | COD$_{Mn}$ | COD | BOD$_5$ | NH$_3^+$-N | NO$_3^-$ | TP |
|---|---|---|---|---|---|---|---|---|
| 6 km | AI.4 [a] | | | | | 0.01 | | |
| | AI.5 [a] | 0.013 | | **−0.070** | −0.030 | | | |
| | PLAND.1 [a] | 0.003 | | **−0.015** | −0.008 | | −0.003 | |
| | ED.1 [a] | 0.023 | | **−0.025** | −0.018 | | **−0.013** | |
| 9 km | ED.3 | | −0.003 | | | | | |
| | AI.3 | | | −0.002 | | | | |
| | AI.4 [a] | **−0.007** | 0.009 | **0.036** | **0.019** | | **0.008** | |
| 12 km | ED.1 [a] | −0.037 | −0.128 | | −0.058 | | | |
| | AI.2 | | 0.0141 | | | | | |

Note: [a] is used to identify the significant independent variables on the basis of VIP (variable influence on projection) scores >1. [b] Bold font indicates the independent variables that passed 95% confidence test ($p < 0.05$), non-bold font represents independent variables that passed 90% significant test ($p < 0.1$).

## 4. Discussion

### 4.1. Seasonal Effects of Water Quality

Wujiang river watershed has a large area (23,000 km$^2$) with diverse landscape settings. Water quality depends on landscape structure, and the nature of relationship between the two is affected by seasonal changes, as depicted in Figures 1–3. Seasonal variation of precipitation and runoff have great effects on stream water contamination concentration [15,38,51]. The concentration of DO is higher during the dry season, while the COD$_{Mn}$ and BOD$_5$ contaminant concentrations are higher during the wet season. DO, COD$_{Mn}$, BOD$_5$, and COD are important indicators of organic matter. They are normally produced by decomposing organic compounds. The more serious the water pollution is, the lower the dissolved oxygen content in water, and the higher the value of COD$_{Mn}$ and BOD$_5$ are [52]. Low DO concentration usually indicates serious aquatic system degradation. Water turbidity

scatters or absorbs sunlight, thereby increasing the concentration of $COD_{Mn}$. As such, $COD_{Mn}$ is higher during the wet season than the dry season [53,54]. In the dry season, water quality can be predicted by class-level metrics within buffer zones of 3, 6, 9, 12, and 15 km, as shown in Tables 2 and 3. In the wet season, water quality can be predicted by class-level metrics only within buffer zones of 6, 9, and 12 km, as shown in Tables 2 and 3. $NO_3^-$ correlated with many landscape metrics during the wet season, which also supports Lee and Liu's conclusion [21,38]. Only organic matter such as DO, $COD_{Mn}$, COD, and $BOD_5$ were correlated, to differing degrees, with landscape metrics in both seasons across all five buffer zones. Water pH and nutrients such as TP and $NH_3^+$-N were weakly correlated with landscape metrics. While cultivated land areas—the main contributors of TP and $NH_3^+$-N—were widespread across the Wujiang river watershed, these two nutrients are easily absorbed and utilized by soil, resulting in insignificant runoff [55,56]. Denitrification occurs under the action of microorganisms. As a result, $NH_3^+$-N is more easily absorbed by soil particles and vegetation than TP and COD [56].

## 4.2. Dominant Landscape Metrics at Class Level Affect Water Quality across Seasons and Buffer Zones

Similar to previous studies [9,21,24,29], certain class-level landscape metrics in this study are significantly correlated to water quality in the Wujiang river watershed across temporal and spatial variables, as shown in Tables 2 and 3. Urbanization increases impervious surfaces which emits both point and non-point pollution into adjacent aquatic systems [13,38,57–60]. PLAND represents the percentage of the landscape that comprises of a particular patch type. AI reflects the degree of dispersion of the patches within the study area, with higher AI values indicating higher aggregation. The present study indicates PLAND and AI of built-up land at the 6 or 12 km buffer zones were positively correlated with COD and $NO_3^-$. Interestingly, AI for built-up land was negatively correlated with TP. Our study supports Liu's conclusion that TP concentrations are likely to decrease when built-up land is scattered [11].

Agricultural areas are a primary source of non-point pollution; conventional agricultural activities result in excessive organic manures, nutrient applications, and inorganic fertilizers, thereby generating pollutants to stream water following surface runoff [59]. PLAND and ED for agriculture land at the 3, 6, and 15 km buffer zone had no correlation, while at the 9 and 12 km buffer zone were negatively correlated to the abundance of organic matter and nutrients during wet season. Other scholars have obtained varying conclusions about the correlation between agricultural land and water quality, due to place-specific differences in topology, terrains, tillage, and fertilization [60]. In the Wujiang river watershed, tributaries and canals are densely distributed in paddy fields, and the paddy fields help purify the water [61]. Xia, Yan, and Yin support the conclusion that multiple paddy field pond systems, which are widely used in southwest China, can reduce agricultural NPS (Non-point Source) pollution and support nutrient and sediment recycling in the terrestrial ecosystem [62,63].

Patch density (PD) and edge density (ED) are effective indicators of landscape fragmentation. High PD and ED are associated with high landscape fragmentation and human disturbance. Forest land cover contributes to water quality [14,15]. AI of forest at the 15 km and ED of forest at the 9 km buffer zone are correlated with degraded water qualities during the dry season. This result supports similar research conclusions that fragmented forests have a low impact on the interception and absorption of NPS compared with dense forests; hence, some pollutants enter the water body through the forest land [64]. AI, PD, and ED for grass were negatively correlated with some water quality indicators at multiple buffer zones during the two seasons. This suggests that high fragmentation of grass can further reduce pollutants. As other studies have also found, grass land plays a retention function to decrease surface runoff and improve water quality [64,65]. There is a strong correlation between AI of water and the amount of organic matter or nutrients during wet season. This suggests the importance of supporting water channel connectivity in the Wujiang watershed. AI of water is the only landscape metric correlated with $NH_3^+$-N, and only in the wet season. This result is understandable due to precipitation and associated runoff. In this study, landscape-level metrics predict most water pollutants only during dry season. ED at the landscape level was negatively correlated with COD, $COD_{Mn}$,

or $BOD_5$ in some buffer zones. Uuemaa, who conducted a watershed study in Estonia, came to a similar conclusion, and explained that this is because complex landscape patterns improve retention of organic matter and nutrients [66]. Liu also found that ED was negatively correlated with contaminant concentrations in agriculture-dominated watersheds in China, suggesting that the strength of the correlation is determined by metrics at the class level [9].

CONTAG measures the extent to which patch types are aggregated. The value of CONTAG is high when high levels of aggregation and low levels of land use type are observed. CONTAG was positively correlated with degraded water within all five buffer zones. This result aligns with other study conclusions that a high fragmentation and low dispersion landscape increases water quality deterioration [14,15,66]. COHESION, which represents physical connectedness of land use within a watershed, was also positively correlated with degraded water quality at the 9 and 15 km buffer zones. This conclusion aligns with previous research, in which degraded water quality is likely to be linked with landscape dispersion [67]. LSI, as an effective indicator to measure the complexity of landscape patch shapes, was negatively correlated to COD, $COD_{Mn}$, and $NO_3^-$. However, LSI was positively correlated with DO within the 3 and 15 km buffer zones. Li and Li have similarly suggested that a complex landscape patch retains water quality [31,68].

## 5. Conclusions

Using GIS tools and PLS regression, this research conducted on the Wujiang river watershed in China's southern karst area yielded the following conclusions:

Water contamination in the Wujiang river watershed is subject to spatial and seasonal variation. Particularly, DO was higher during the dry season, while other contaminants were higher in the wet season. PLSR analysis showed a definite correlation between landscape structure and water quality, varying according to changes in rainfall. During the dry season, landscape metrics at class and landscape levels predicted water contaminants, especially the presence of organic matter. This was true across all buffer zones. During the wet season, only metrics at the landscape level were correlated with organic matter and nutrients. This was true for buffer zones at the 6 to 12 km scales, but not at 3 or 15 km scales.

These conclusions show that watershed buffer areas with fragmented, small-sized patches of crop land and built-up land and high aggregation of forest land are correlated with better water quality. Degraded water quality is associated with highly aggregated patches and had a high degree of aggregation and good physical connectedness. Water pollution mitigation measures in the Wujiang river watershed may include reforestation measures and the maintenance of connectivity between tributaries.

Our study helps fill gaps in the literature regarding exploring water quality across a large area of typical karst landscape from the perspective of landscape ecology. GIS and statistical analysis are used to understand the distribution characteristics of water quality in the watershed, and analyze the relationship between water quality and landscape structure. The results of this study can help inform specific water protection and monitoring mechanisms for the Wujiang river watershed, and offer a valuable reference for research on the relationship between water quality and land use in karst areas.

**Author Contributions:** G.X. and X.R. conceived and designed the experiments; G.X. and Z.Y. performed the experiments and collected data; G.X. analyzed the data; J.X. contributed reagents/materials/analysis tools; G.X. and X.R. wrote the paper; H.L. helped to provide the part of the data in this article.

**Funding:** This work was supported by Project of National Key Research and Development Program of China in the 13th Five-year Plan: Ecological industry model and integrated technology and demonstration for the rocky desertification control of the karst plateau gorge [2016YFC0502607]. G.X. was supported by the Project of Innovation Program for Postgraduate Education of Guizhou Province: Xiong Kangning's studio of postgraduate supervisors for the karst environment of Guizhou Province (04 2016 Qianjiao Yanhe GZS ZI) and Science Foundation for Doctorate Research of Guizhou Normal University (No. 119040516016). National Natural Science Foundation of China "Study on the Law of Soil Erosion and Its Coupling with Rocky Desertification in Typical Karst Areas" (No. 41561066).

**Acknowledgments:** We also thank B.H. (Macau university of science and technology, Department of Decision Sciences, School of Business) for helping with the statistical analysis.

**Conflicts of Interest:** The authors declare no conflict of interest.

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
