# Peer review of "Influence of Landscape Structures on Water Quality at Multiple Temporal and Spatial Scales: A Case Study of Wujiang River Watershed in Guizhou"

_water, doi:10.3390/w11010159_

Round 1

Reviewer 1 Report

Dear Authors,

While trying to review the manuscript, I did not find either page number or line number. Furthermore, excessive grammatical flaws made the manuscript too hard to follow. Hence, I can only provide comments once you address these two issues.

Regards,

Your Reviewer 

Author Response

dear reviewer,

I have checked the grammar mistakes and change the sentence formation in my article,thank you for your advice.

Reviewer 2 Report

Please see the attached pdf file for detailed comments

Author Response

Response to Reviewer 1 Comments

Point 1: The whole abstract is poorly written. First define the water quality problem, then effect of different landscapes on water quality. What water quality variables were monitored, Is organic matter a contaminant.

Response 1:  I haved rewritten the whole abstract in  line 13 to 32. Landscape structures lead to water quality degradation, although its influence is seasonal and spatial dependent. In our study, we took Wujiang river watershed as an example to assess the relationship between landscape metrics and river water quality. Such an assessment can greatly improve water contamination predictability and provide restoration strategies and management suggestions. Eight water quality variables including PH were collected from 32 sampling sites from 2014 to 2017. NH3-N, NO3, TP were selected to represent nutrient pollutants. COD, BOD5, DO, CODMn were representative water quality variables for organic matters of oxygen consumption. PLS regression was integrated to analyze quantitatively the influence of landscape patterns on water quality at five buffer zone scales (3, 6, 9, 12, and 15 km) in Wujiang river watershed. Results revealed that water quality had different responses with landscape metrics due to precipitation. Landscape metrics at class level were effectively predicted water quality degradation in comparison with metrics at landscape level. During dry season, the landscape metrics at landscape and class levels had predicted organic matters at the five buffer zone scales. However during wet season, only the landscape metrics at class level had predicted water contaminants, such as organic matters and nutrients, at three buffer scales. We identified the following important indicators of water quality degradation: percent of landscape, edge density, and aggregation index for built-up land; aggregation index for water; contagion; CONSHION; and landscape shape index. Water pollution can be probably mitigated by reducing natural landscape composition fragmentation, increasing the connectedness of region rivers, and optimizing human disturbance landscape structures in karst areas.  

Point 2: Rewrite sentence. It is scarcity of surface water or fresh water.

Response 2: I have rewriten the senstence in line 37 to 39. It is the scarcity of surface water and deterioration of water environment that  are globally crucial environment issues along with rapid urban sprawl and economic development [4-5].

Point 3: English is badly written. Define what is IGBP

Response 3: Sorry for forgreting explain what is IGBP. I have rewritten the sentence in Line 40 to 43. As IGBP (International Geosphere-Biosphere Program) and LUCC projects are implemented worldwide, research on water environment related to the change of landscape pattern has become a trending topic in the field of resource and environment science

Point 4: instead of through intervening-use affecting

Response 4: I have rewritten the sentence in line 45 to 46. Water quality causes changes in land use and landscape pattern by intervening hydrological processes and ecological systems

Point 5: What is "researches" do you mean to use the word "researchers"

Response 5: However, a discrepancy emerges in existing research regarding which aspect of landscape pattern characteristics should be analyzed. These studies on land use factors or landscape metrics at landscape level have used different methods.

Point 6: Do mean "Independent"

Response 5: I have rewritten the sentence in line 78.

Point 7: Noise dates? do you mean data

Response 7: yes, noise date are used in other articles and a book named an introduction to statistical learning to described why PLS regression was used.

Point 8:owns-use has

Response 8: I have rewriten the whole sentence in line 90 to 91.Different watersheds have various characteristics such as land cover, topography, climate, soil, and geology

Point 9: Write like this-In event of the drought, the river runs dry i.e. no water flow

Response 9: I have rewritten the sentence in line 98 to 99. During seasonal drought, some rivers run dry due to lack of water supply.

Point 10:What is the meaning of the word "Feather"

Response 10: the whole sentence was changed into The shape of Wujiang river system is like a feather. in line 110.

Point 11:use the word dependent instead of fed

Response 11: I have changed fed into dependent  in line 114.

Point 12:write like this-but some areas are highly urbanized.

Response 12: I have changed fed into dependent  in line 114.

Point 13:write like this-but some areas are highly urbanized.

Response 13: I have rewriten the whole sentence in  line 116 to 117. However, some regions have a high urbanization rate; for example, Yunyuan district has an urbanization rate of 97.31%.However, some regions have a high urbanization rate

Point 14:Urban life is the source of pollution-urban wastewater is the source of pollution and not urban life. Please be careful while using words.

Response 14: Urban life is the source of pollution-urban wastewater is the source of pollution and not urban life

Point 15:What do you mean by detection section?

Response 15: I have already changed the sentence into The water quality detection section was set up in the inlet and outlet of the upstream tributaries in line 128 to 129.

Point 16:Is monthly sampling enough to describe temporal and spatial trends in water quality? I think there is a need to increase the sampling frequency.

Response 16: All the experiment  was conducted as GB3838 (the environmental quality standard for surface water )

Point 17:Corp land-do you mean crop land?

Response 17: Corp land means arable land, but in the field of GIS, It is represented 6 major type of land.

Point 18: Improper English grammar. It can be written as Prior small-scale (<3km ) buffer studies explored the correlation between water quality and landscape metrics, with land and water quality affected by human interventions.

Response 18: Thank your so much for your advice. I have changed the sentence  Prior small-scale (<3km ) buffer studies explored the correlation between water quality and landscape metrics, with land and water quality affected by human interventions. 

Point 19: Improve the English-how their responses to each other? what do you mean by this sentence.

Response 19: sorry to confuse you . I have changed the whole sentence into  PLS models for the eight water contaminants and 33 landscape metrics were constructed to identify their responses to one another in line 188 to 189.

Point 20: Say  log transformed

Response 20: I have change the sentence into All the variables were log transformed  to prevent negative values of predictions and to normalize the distribution of each variables in line 193 to 194.

Point 21:for explaining the independent variable.

Response 21: sorry for making the mistake, I have already changed  for explaining the independent variable  into for explaining the independent variable.

Point 22:write like this-highly affected by human activity were of the inferior grade V.

Response 22: I have changed the whole sentence in line 215 to 216. However, the water quality parameters in six sampling sites were considerably affected by human activity. They met the standard of Grade inferior V.

Point 23:Use of the English language has to be improved drastically-sentence formation is not correct. Very difficult to understand.

Response 23:Thank you for your suggestion  I have checked the grammer mistakes in line 360 to 385. If you still confuse by the sentence I will conduct another English editing.

Wujiang river watershed has a large area (23,000 km2) with diverse landscape settings. Furthermore, its water quality parameters have spatial and seasonal variability, as depicted in Figs. 1, 2, and 3. Seasonal variation of precipitation and runoff have great effects on stream water contamination concentration [15,38,51]. Figs. 2 and 3 reveals spatial and seasonal variations in average water contaminants. The concentration of DO is lower during wet season than those during dry season. The CODMn and BOD5 contaminant concentrations display an opposite trend. DO, CODMn, BOD5, and COD are important indicators of organic matters. They are normally produced by decomposing organic compounds. The more serious the water pollution is, the lower the dissolved oxygen content in water, and the higher value of CODMn and BOD5 are [52]. Moreover, low DO concentration usually indicates serious aquatic system degradation. Water turbidity makes the light scattered or absorbed, thereby affecting the determination of the amount of CODMn; as a result, the concentration of CODMn is higher during wet season than that during dry season [53]. Research on the relationship between landscape metrics and water quality has been conducted on both seasons. However, compared with the dry season, response variables in the wet season can be further predicted by landscape at class level within the three buffer zones (Tables 2 and 3). This result is consistent with previous studies where pollution is associated with land uses [53,54]. NO3- correlated with many landscape metrics during wet season also support these conclusion. Nonetheless, only organic matters such as DO, CODMn, COD, and BOD5 are strongly correlated with landscape metrics in both seasons in the five buffer zones, although the correlation was different. Nutrients such as TP and NH3+-N have weak correlation with landscape metrics. This result is possibly because the cultivated land areas is large in Wujiang river watershed. Furthermore, the surface erosion is relatively weak; hence, TP is easily absorbed and utilized by soil in the long-term runoff transportation and movement, thereby resulting in the insignificant impact on landscape metrics during both seasons [54]. Wang conducted a research in Ganjiang river in China and obtained the same conclusion. NH3+-N is more easily absorbed by soil particles and vegetation than TP and COD are. Denitrification occurs under the action of microorganisms [55].

Point 24:Please write proper English--please Positively charged ammonia nitrogen was more easily absorbed by soil particles and vegetation, and the denitrification occurred under the action of microorganisms

The sentence in Line 384 to line 385 was changed into NH3+-N is more easily absorbed by soil particles and vegetation than TP and COD are.

Point 25:This study demonstrates the relationship between land use, land type and resulting water quality parameters affected by human interventions. This is really a good technical study, but due to poor sentence formation, it is difficult to read.

I have rewriten the whole paragraph from 445 to 465 : Using GIS tool and PLS regression, the research conducted on Wujiang river watershed in southern karst area China yielded the following conclusions:

Water contamination in Wujiang river watershed presented spatial and seasonal differences. Particularly, DO was high during dry season, and other contaminants showed an opposite trend. The relationship between landscape structure and water quality were affected by rainfall. More water quality variables were predicted with landscape metrics though only at class level during wet season than during dry season within 3 buffer zones. However during dry season, landscape metrics at the class and landscape levels predicted water contaminants, especially organic matters, within the five buffer zones, although their influence was different.

Following PLSR, the RC and the VIP value illustrated the different landscape metrics had different scale effects. Among the class-level configuration metrics, PLAND, ED for crop land, PLAND, ED, PD for built-up land, PD, ED for grass, and AI for water had different contributions to water degradation during dry and wet seasons. The majority of landscape-level configuration metrics, such as PD, ED, LSI, CONTAG, and CONSHION, presented different correlations with degraded water quality particularly during dry season. Better water quality was present in watershed that contained landscape fragmentation of crop land and built-up land with small patch size and high AI. Water pollution can be probably mitigated with non-fragmented forests and maintained with the connectedness of rivers. Degraded water quality connected with patches had a high degree of aggregation and good physical connectedness. Thus, land use planning should optimize landscape composition and landscape configuration and adopt a multi-scale perspective. This recommendation is an alternative strategy that can alleviate water contaminants.

Reviewer 3 Report

This paper explores the impact of land use on water quality, which is a topic of interest to a range of audiences.  What is not clear from the paper is how this study makes a contribution to the literature.  It is mentioned that there has not been work done previously in karst areas,and the study may have local interest, but it is not clear how that would contribute to the field more generally.  

I have several broader issues that I think need to be addressed:

The writing needs significant review to ensure communication is clear.  Connected to this is the need to use clear phrasing e.g. the use of the term landscape when referring to both the metrics and the scale of analysis is confusing. You also have metrics that measure distribution, so the intent when you are talking about distribution of the means of these metrics in your results is unclear.

The study looks at 5 different buffer zone sizes and while this is looking at different spatial extents, I'm not sure it is truly a multi-scale analysis.  This does not diminish the value of considering the different extents, which is a research question explored by many, but I'm not sure it is truly a multi-scale analysis.

Table 1 doesn't seem to include all the variables discussed in the text (LPI, FRACT, CONSHIN or CONSHOIN--you use both spellings, I'm not sure which is correct).  The table could have a column to communicate whether something is landscape, class or both.  Variables in the table need to be defined. You later talk about SHDI and CONSHION in section 3.4, but I don't see either of those in Figure 6.

Your presentation of PLSR is confusing.  I think you have some of the terminology reversed (though I think this is a communication issue, not a technical one). You don't really clarify how you use cross-validation in here (leave X out?)

Figure 2 suggests to me that the early and late dry seasons are quite different for some variables (e.g. DO, TP, CODMn).  Bundling these together seems like it might impact your analysis and discussion of correlations to the landscape metrics.

My understanding of Figure 4 is that I want to use it to look at the different sites and how the land use around them varies across the different spatial extents.  I think this is hard to interpret from the current layout.

Your presentation of the results related to the landscape metrics across the different scales is confusing. Beyond this, you seem to calculate averages across the 32 sites and then focus on distribution of the mean values.  Since the land use is different, I'm not sure how taking an average across the sites makes sense.  This is a fundamental element of the paper so needs to be clarified.

Your results (e.g. section 3.5) duplicates information that is found in figures or tables.  This increases length and in some cases, decreases clarity.

Your discussion does a good job of pulling in the literature, but the communication would be enhanced with some editing.

I have a large number of suggested edits and specific comments.  I will include those in a marked up document rather than try to replicate them here.

Author Response

Response to Reviewer 2 Comments

Point 1: English editing

Response 1: Thank you so much , I  have found a English native speaker to help do the English editing face to face according to your request. I have highlighted the modification part I had made in blue color.

Point 2: line 23, Abbreviations generally must be spelled out before use; I'm not sure if chemical names necessarily fall under that umbrella, but the other abbreviations (e.g. COD, BOD, DO, PLS, CONSHION) would need to be spelled out.  In the case of the first group of these, the bigger question is if they are needed in the abstract or is stating the general categories sufficient.

Response 2: Thank you so much for your suggestion. I have already rewritten the whole abstract. Line 21 to 24. Including add the full name of the water quality pollution.

Point 3: line 24, I'm not sure what you mean by class level?  You later define this by talking about land use classes, but I think "land use" needs to be added in the abstract when you talk about class.Phrasing is confusing here.

Response 3: I delete the sentence in my paper, and rewritten the in order not to confuse the reviewer and reader. Line 27 to 28. “Results revealed that water quality is affected by landscape composition, landscape configuration, and precipitation. During dry season, landscape metrics at both landscape and class levels predicted organic matters at the five buffer zone scales. ”

Point 4:line 27 What about the other two buffers?

Response 4: I have rewritten the whole sentence in line 31. During wet season, only class-level landscape metrics predicted water contaminants, including organic matters and nutrients, at the middle three of five buffer scales.

Point 5: line 30 Based on your results?

Response 5 : I have rewritten my sentence in line 34 to 36. “These results suggest that pollution can be mitigated by reducing natural landscape composition fragmentation, increasing the connectedness of region rivers, and minimizing human disturbance of landscape structures in the watershed area.

Point 6: line 37 to line 39. It is the scarcity of surface water and deterioration of water environment that are globally crucial environment issues along with rapid urban sprawl and economic development

Response 5: I have rewritten the sentence in line 41 to 42.“Urbanization and economic development have put significant stress on water systems across the world, making water quality an issue of global importance

Point 6: line 40 to 43. As IGBP (International Geosphere-Biosphere Program) and LUCC projects are implemented worldwide,research on water environment related to the change of landscape pattern has become a trending topic in the field of resource and environment science

Response 6: line  44 to 47 As International Geosphere-Biosphere Program (IGBP ) and Land-Use and Land-Cover Change (LUCC) projects are implemented worldwide, research on the effects of changes in landscape patterns on the water environments has become a trending topic in resource management and environmental science fields.”

Point 7: line 45 to 46. Is this back-to-front?  Water quality causes changes in land use (possibly) and landscape  patterns?  Or changes in land use and landscape patterns change water quality?

Response 7: Thank you for your suggestion. land use and landscape patterns affecte water quality. I haven’t explain my view clearly. I have rewritten the two sentence in line 47 to 48. In addition, land use and landscape structures also affect hydrological processes and water quality.The relationship between water quality of rivers and riparian land use are subject to spatiotemporal variation.”

Point 8:line 47? Phrasing is awkward.

Response 8: I have deleted the sentence in my article.

Point 9: Line 48,Is the water quality of river systems impacted by spatial scale or is our observation of that phenomena impacted by scale?   

Response 9: We explore the relationship between water quality and landscape structures by different buffers scales. Because different buffers had different landscape structures.

Point 10:  But not quantifying?  Is that your point here?

Response 10: sorry to confuse you, Both quantitative and qualitative analysis. I have changed the whole sentence into  the development of GIS has helped advanced both quantitative and qualitative analysis tools for understanding the relationship between land structure and water quality, and have contributed greatly to watershed planning and management fields. in line 51 to 54.

Point 9: line 50 to 52 ?Meaning it is hard to compare studies?The first half of this sentence isn't connected to the second half.Understanding the relationships is important for planning, and perhaps the development of methods helps to describe these relationships, but your phrasing is not quite right as is.

Response 9: the sentence I have written haven’t send right information, So I deleted it. I have rewritten the second sentence “The development of GIS has helped advanced both quantitative and qualitative analysis tools for understanding the relationship between land structure and water quality, and have contributed greatly to watershed planning and management fields” in line 51 to 53.

Point 10: line 53 to 54 .Word choice?

Response 10: line 55 I have rewritten the whole sentence in “Landscape structures are based on land use composition and landscape configuration”.

Point 12: line 54 .How do you define a class?  (You later define this as a land-use class, but I think you need to be clearer up front that is what you are talking about.

Response 12: sorry to confuse you. Class-level metrics describe the amount and spatial distribution of patches belonging to a particular land use class; I have explained it in line 57 to 59.

Point 13: in line 59. Are the citations you give examples of studies that performed class-level analysis?  If so "few" doesn't seem like the right word.  Some seem to be, but others are landscape level studies.  This is confusing and is important because I think it is a justification of your study.

Response 13: Most of the article I cited in sentence are conducted on the landscape level, very few of study conducted this study on class level.I had changed the sentence in line 62 to 64.

Point 14: line 62, I'm not sure I understand the use of "response" here. A medium for what?Awkward phrasing.

Response 14: I  have deleted the whole sentence and rewritten these sentences in line  67 to 70  Buffers zones can greatly mitigate denitrification and the deposition and absorption of contaminants from non-point pollution into adjacent water. The mitigation effects of the buffer zone vary based on its proximity to the water source. As such, the relationship between water quality indicators and landscape metrics are scale-sensitive [26,27-28]. Studies have highlighted land use/landscape within different buffers zones scales.

Point 15: In line 72 on water quality or something else?

Response 15 : I have changed the whole sentence in line 76 to 78  Li found that riparian land use exerted the largest influence on the upper Han River basin of China during the dry season. 

Point 16: Line 79 to 80 . into what?

Response: I had changed the whole sentence into Landscape metrics are neither independent variables nor do they exist collinearly. Some indicators can skew statistical analysis and yield potentially misleading results because different landscape indicators may be interrelated. see line 82 to 84

Point 17: line 80 to 81, What does that mean?

Response: I had changed the whole sentence in line 85 to 86. High multicollinearity will cause a poor performance of multivariate regression even if a subset of expression levels is selected .

Point 18: Line 80 .What does that mean? Noisy data

Response:I deleted the noisy data. I have changed the whole sentence into “High multicollinearity will cause a poor performance of multivariate regression even if a subset of expression levels is selected “see  line 84 to 85.

Point 19: Line 83 You used this abbreviation in your abstract but have never spelled it out. Separately, you need a transition between the prior sentence and the discussion of PLSR.

Response: I have spell it out in line 88, see the word Partial least square regression (PLSR) helps overcome this issue by combining the features of principal component analysis and multiple linear regression.

Point 20: line 91. The prior sentence doesn't lead to the next statement, so this isn't the right transition.  Perhaps "however" would be better.

Response: Sorry to confuse you. I have rewritten the whole sentence.The relationship between the landscape characteristics of a watershed—including land cover, topography, climate, soil, and geology— and water quality is complicated and place-specific [44-45]  see line 94 to 96.

Point 21: What studies? Line 92 . Source? Line 94.  You can say "no study" or "few studies", but not "nearly no study".What is a karst area?  You haven't talked about this at all yet. Line 94.

Response: I have already cite the literature in line 98. I have rewritten the senstence “While many studies on the influence of landscape structure on regional hydrological processes and water quality have been conducted, there has only been one published study on large watersheds in karst areas (published in Chinese)”. I have introduced what is special characteristics of karst areas in line 98 to 100.Karst areas, due to their steep slopes and shallow, highly erodible soil, create particularly vulnerable conditions for watersheds

Point 22: Line 110. This doesn't look particularly west in your graphic.  

Response 22: But Gui zhou province is located in Southwest in China.

Point 23: It at the start of the sentence was talking about the watershed, but I presume here you mean the river?  I generally avoid "it" so such confusion is less likely.

Response 23: I have changed the whole sentence into The watershed covers a total area of 32,300 km2, and runs 889 km in length from west to east.  in line 112 to 113.

Point 24: Line 111. I'm not sure what this means or what this adds.

Response 24: I have deleted  this senstence.

Point 25 : line 116 percentage isn't a rate. Do you mean proportion?

Reponse 25: I have changed the whole sentences into “The population of the Wujiang river basin in 2015 was over 21 million, with 61.3% of the population involved in agricultural production. 37.47% of the population in the Wujiang river watershed lives in urban areas and are involved in non-agricultural livelihoods. “line 115 to 118.

Point 26: Line 118 .Are you saying the Yunyuan district is 97.31% urbanized?  Phrasing is awkward here.

Response: I had change the whole sentence in line 118. Some districts have a high urbanization rate; for example, Yunyuan district has an urbanization rate of 97.31%. in line 118 to 119.

Point 27: Line 118. Which area?  The Province? The watershed? the Yunyuan district? 

Response: line 118, yes, Yunyan district Mining and manufacturing industries are dominant in this district. Watershed pollution sources include industry, urban waste water, agriculture, and large-scale livestock farming. In line 119 to 121.

Point 28: Line 120 to 121, I'm not sure what this means and if it is particular to your study area.  Perhaps just phrasing.

Response: Sorry to confuse your I  deleted this sentence.

Point 29 :line 124 I'm not sure the buffer insets are needed.  I think you'd be better to remove these and then you could plot both parts at the same scale, which might make interpretation easier.

Response 29 : I  have deleted one of the insets that might confuse you.

Point 30: Line 127 to 128,Why not the others?

Response: I have change the whole sentence into “32 sample sites along the main stream and its tributaries were established by the Guizhou Environmental Monitoring Centre to monitor water contaminants in the watershed.” consider the complete of data, 32 sampling sites was chosen as study area. See  line  126 to 128.

Point 31 : Line 136 , line 137. This is hard to follow in the text.  Could this go into a table? You could have a column in the table for variable type (e.g. nutrient pollutant), name (e.g. total phosphate), abbreviation (e.g. TP), and measurement method.

Response:  Thank you for your suggestion, I have spell all the abbreviation in the phrasing. And make a table in my article. see table 2.

Point 32: Line  137. You don't mention pH above (and the p should be lower case) so this doesn't make sense.

Response: because pH have  not going into the PLS regression model.

Point 33: Line 158. 15?

Response:I have changed the whole sentence into Most sample sites in our study had large catchments, so a maximum of 15 km was used as the buffer zone.  see line 164 to 165.

Point 34: line 161 to 165. Now you define it.  Even if all you do earlier is specifically say land use class that would make this clearer.

Reponse: Thank you for your suggestion. I had explained the metrics at class level and at landscape level in the introduction. See line 56 to 59.

Point 35: line 164. It is confusing to have landscape metrics at the landscape scale AND landscape metrics at the class scale. Line 165. If you add a column to the table, you could probably eliminate these three sentences. Line 168.

Reponse:  Sorry to confuse you, I had explained the difference between them in line 56 to 59.

I had deleted these sentences and add the missing information about metrics on the table.

Point 36: line 165. Table 1: You need to define all your variables somewhere.  Some are described in the description, but not matched to the symbols.And I think you need to pull the footnotes out since you seem to have a used in some equations as well as the footnote.  You could all another column for scale and put in Class, Landscape or Both. Since I don't see 4 class level metrics indicated here.Are these equations all as defined and applied in FRAGSTATS?

Response 36: I have supply all the missing information about metrics both at class level and landscape level on table 2. Yes,  they are all  defined and applied in FRAGSTATS.

Point 37: line 182. You have only referred to Figure 1 so far, so can't refer to figure 6.  In any case, if this is from the results, don't mention it here.  And I think you meant figure 7 anyway.

Response 37:  Sorry for  confuse you. I have deleted the word “Fig 6” in line 183.

Point 38: line 182 Meaning what?  Or is that what comes in the next paragraph?  If so,wouldn't break them up.

Response 38: I had changed it in line 184 to 185.  PLSR was selected to analyze the impact of landscape metrics on water quality across different buffer zones and seasons.

Point 39: line 184 to 185. But how is this different to any other type of regression? And I thought X was the predictor variable not the predicted variable.  Abdi says "GOAL OF PLS REGRESSION: PREDICT Y FROM X".  I think your phrasing is off here.

Response 39: PLSR can handle highly correlated variables by explicitly assuming dependency among the variables and estimating the underlying structures and is particularly suitable for cases in which the number of observations is less than the possible variables (Luedeling and Gassner,2012 Luedeling, E., Gassner, A., 2012. Partial least squares regression for analyzing walnut phenology in California. Agr. For. Meteorol. 158–159, 43–52.). I have changed the sentence into PLS finds a set of orthogonal X-components that maximizes the level of explanation of Y. This method also provides a predictive equation for Y in terms of the X’s. ” line 185 to 186.

Point 40: line 191 to 192.change the predictor into response.

Response 40:  The whole sentence have changed into “PLS models for the eight water contaminants (Y) and 33 landscape metrics (X) were constructed and analyzed for correlation in line 188 to 189.

Point 41: What do you mean fully cross-validation?

Response 41: sorry to confuse you.I have change the whole word into “The appropriate number of significant PLSR components was determined using cross-validation. in line 191.

Point 42: line 197, Why this value?

Response 42: the Cumulative contribution Frequency is up to 60% in considered the model has a fit of goodness.

Point 43: line 200, delete VIP,  line 203, delete one. Add al

Response 43 : I have  deleted “ VIP” in line 197, deleted one in 200. I had changed the word statistic into statistical.

Point 44 : line 210. What independent samples?  

Response 44: sorry to confuse you. I have change the whole sentence into “The t-test indicated that water quality indicators in most sampling sites exhibited significant seasonal differences (P<0.05) in line 198 to 199.

Point 45 : line 212. delete “in ArcGIS 10.2 in the 32 sampling sites.”

Response 45: I have changed the word into”The mean concentration of the eight water contaminant parameters was mapped using ArcGIS 10.2. Natural break method was adopted to group the contaminant concentrations into five categories.”see line 180 to 181.

Point 46 : line 214. I don't like this word, particularly because I don't think this patterns are obvious. Some of the variables have seasonal differences, but others are pretty flat and the two pieces of the dry season are markedly different in places.

Response 46: I deleted the word “obvious”. sorry to confuse you, because we use the average  date of 32 sampling sites to deplicted the data existed seasonal difference. The concentration of water pollutants in some site are high, but in some site are low. The t-test indicated that water quality indicators in most sampling sites exhibited significant seasonal differences. In the field of this study, most of researchers divide just wet and dry season.see the reference: bu(2014 ); shen(2014), shen(2015)

Point 47:  Perhaps that could be highlighted in Figure 3?

Response 47: I think it is hard to highlight in Figure 3. And it is not an important information in my article.line 226.

Point 48: line 218, What is this?

Response 48 : (GB3838-2002) is Environmental quality standards for surface water in line 214.

Point 49:  line 219 Which are the major sites?

Response 49: I mean the majority in Line 215.

Point 50: I agree there is more variation across the year, but the differences between the pieces of the dry season are just as different as they are to the wet season in many cases.

Response 50: In the field of this study, most of researchers divide just wet and dry season.see the reference: bu(2014 ); shen(2014), shen(2015)

Point 51 : line 228,  The X-axis is each month of the year? I would expect month 1 and month 12 to be similar, but this is not at all true for TP.  Is there a reason for that?

Response 51 : I had add “month” in fig 3. The range of TP range from 0.11 to 0.23. Actually the difference is quite small.

Point 52: line 231. Figure 2 I'm not sure I find this figure really useful.  The different dot sizes are hard to interpret, but I'm also not sure I have a better alternative to suggest.  Doing this as a table would be more difficult to interpret I expect.

Response 52: yes, I agree with you.it is hard to illustration the spatial distribution of sampling sites.

Point 53: line 233 , line 241, line 242. line 245. Significant has statistical connotations.  Is that what you mean?I'm not sure that's the right phrasing.Structure here is not clear.

Response 53: I have rewritten the whole part of “3.2 Characteristics of landscape composition 233 at multiple scales” in line 229 to 242.

Point 54: line 249. I'm not sure if it is feasible to display these by site, not buffer zone.  What I really want to do with this figure (I think) is compare the proportions of cover types for a particular site across the 5 buffer zones.  I have a hard time doing that here.  If you had 32 parts, each with five bars, I think you could interpret this more easily.

Response 54 :I think this figure in the easiest way to illustrate. The land cover  on  32 sampling site  on five buffer zones. 

Point 55: Line 251 to 263

Response 55: I have rewritten the who part 3.3 Characteristics of landscapes at class level at multiple scales in line 247 to 263.

Point 56:  line 266 to 292.

Response 56: I have rewritten the who part 3.4 Characteristics of landscapes at landscape level across buffer zones

Point 57: line 307 to 308 .If this is in the table (here and throughout the rest of the paragraph), don't duplicate it in the text.

Response 57: Thank you for your suggestion, I have rewritten the whole part “3.5 Linkages between water contaminants and landscape metrics across buffer zone scales based on PLSR “ in line 287 to 313.

Point 58: line 373, You mean that's what you focused on or what previous research has addressed?  Passive voice makes this harder to interpret. 

Response 58:  I have rewritten the whole part  4.1 Seasonal effects of water quality in line 334 to 354 line. I have deleted this word in line 373 to 374Research on the relationship between landscape metrics and water quality has been conducted on both seasons., which confused you and readers.

Point 59: line 375. You need to find an alternate phrasing since you have landscape level and class level, to talk about the landscape at class level is confusing. And you are talking about the class-level metrics right? in Line 375.

Response : Sorry to confuse you. I just want describe the water quality correlated with landscape structure. Poor English written convey wrong message, I have changed the whole sentence into Only organic matters such as DO, CODMn, COD, and BOD5 were correlated, to differing degrees, with landscape metrics in both seasons across all five buffer zones. Water pH and nutrients such as TP and NH3+-N were weakly correlated with landscape metrics.in Line 348 to 350.  

Point 60: line 391, I didn't get from the prior sentence(s) that you thought this was an inconclusive result; Line 403 Spell out.  Line 413. Sentence is a fragment; And you still need a word after this e.g. pollution

Response 60: I have rewritten the whole part ” 4.2 Dominant landscape metrics at class level effect water quality across seasons and buffer zone. Sorry to confuse you, I have rewritten the whole sentence:” Other scholars have obtained varying conclusions about the correlation between agriculture land and water quality, due to place-specific differences in topology, terrains, tillage, and fertilization” in line 347 to 410.

Point 61: line 457, as a conclusion I'd like to know what kind of contribution it was.

Response : Thank you for your suggestion. I have  supplid the contribution of my study on line 429 to 435.

Point 62 :Since you bundle all the observation points together it doesn't seem like you have the data to support this statement. Do you have sufficient data to group the points by their land use characteristics?

Response : we can  make a clustering analysis  to group the points by land use characteristics. But I dont think it necessary to do that. Because We use the data for landscape structures in 32 sampling sites to conducted the PLSR models which can enough to get the correlations between water quality and landscapesture.

Round 2

Reviewer 1 Report

While the authors claimed that they addressed the grammatical issues, but I did not see any evidences of it. As also pointed out by other reviewer, it is very difficult to review the article as such. Just to show examples, I have pointed out some mistakes in the Abstract:

Line 12: “spatial” should be changed to “spatially”

Line 12: A better word for “took” would be “considered”

Line 20: “PLS” – full form for the first time

Line 21: Numeric values of 3,6,9,12 and 15: distance from where? Not clear!

Line 22: “landscape matrices”: what are them? Authors never introduced them and suddenly it appears in the results.

Line 29: “CONSHION” – is this full form for something? I don’t know if it is English word.

Line 33: “PLSR” – never used in abstract and suddenly appears here.

Hence, in general, it is very hard to review the manuscript. The manuscript might have some good insights and contribution to hydrological and water quality modelling realm, however, I can’t review it unless I can understand what authors are trying to say.

Author Response

Thank you for your suggestion. I have got a second round English editing face to face with a  English native speaker. 

Reviewer 3 Report

The authors have fundamentally addressed my concerns.  Those that remain are preference to some degree, which I am OK with.   The writing is substantially improved, which has clarified many sections of the document e.g. the introduction and discussion are far stronger now.  The document needs a final edit, e.g. there are still undefined acronyms used, but this is of a relatively minor level.